# 👻 FANToM:
# A Benchmark for Stress-testing Machine Theory of Mind in Interactions

**Hyunwoo Kim**[♡]  **Melanie Sclar**[♠]  **Xuhui Zhou**[◇]
**Ronan Le Bras**[♡]  **Gunhee Kim**[♣]  **Yejin Choi**[♡♠]  **Maarten Sap**[♡◇]

♡ Allen Institute for Artificial Intelligence  ♠ University of Washington
◇ Carnegie Mellon University  ♣ Seoul National University

## Abstract

*Theory of mind* (ToM) evaluations currently focus on testing models using passive narratives that inherently lack interactivity. We introduce 👻 FANToM, a new benchmark designed to stress-test ToM within information-asymmetric conversational contexts via question answering. Our benchmark draws upon important theoretical requisites from psychology and necessary empirical considerations when evaluating large language models (LLMs). In particular, we formulate multiple types of questions that demand the same underlying reasoning to identify *illusory* or false sense of ToM capabilities in LLMs. We show that FANToM is challenging for state-of-the-art LLMs, which perform significantly worse than humans even with chain-of-thought reasoning or fine-tuning.[1]

## 1 Introduction

Existing evaluations for language models' *theory of mind* (ToM) – i.e., the ability to understand the mental states (e.g., thoughts, beliefs, and intentions) of others (Premack and Woodruff, 1978), is primarily focused on using situation descriptions (i.e., narratives) as the target domain (Nematzadeh et al., 2018; Le et al., 2019; Sap et al., 2022; Shapira et al., 2023a). However, ToM capabilities play an even more important role in understanding dynamic social interactions, as they form a crucial component of effective communication (Frith, 1994; Schober, 2005). Furthermore, as narratives condense situation information into short texts, reporting biases can cause them to include spurious correlations or surface cues (Gordon and Van Durme, 2013). These can be exploited by large language models (LLMs) to display *illusory ToM* – i.e., a false sense of robust social reasoning by models.[2]

In this work, we introduce 👻 FANToM, an English benchmark **f**or stress-testing m**a**chi**n**e **ToM**

---

[1] https://hyunw.kim/fantom

[2] We do not believe that current LLMs possess an actual ToM. Please see §8 for further discussions.

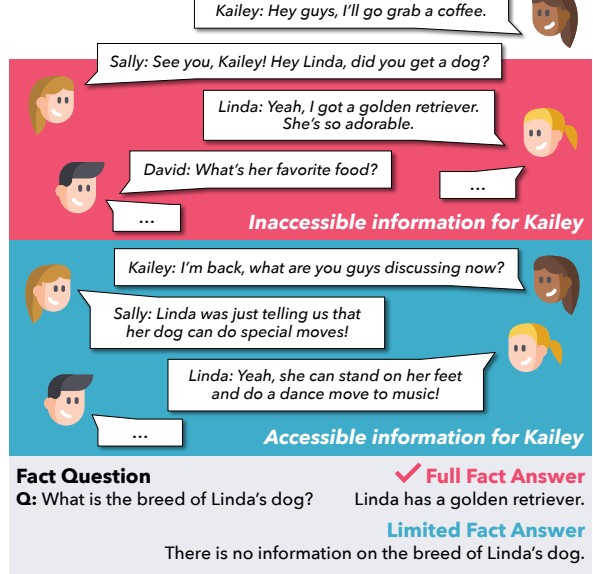

**Fact Question**
**Q:** What is the breed of Linda's dog?

✓ **Full Fact Answer**
Linda has a golden retriever.

**Limited Fact Answer**
There is no information on the breed of Linda's dog.

**Theory of Mind Questions**

• **Belief Question**
**Q:** What breed would Kailey think Linda's dog is?

**Omniscient-view Belief**
Kailey believes Linda has a golden retriever.

✓ **Kailey-centric Belief**
Kailey does not know the breed.

• **Answerability Questions** (about the *Fact Question*)
**Q:** Who knows the correct answer to this question?
**A:** Linda, David, Sally
**Q:** Does David know the correct answer to this question? **A:** Yes

• **Info Accessibility Questions** (about the *Full Fact Answer*)
**Q:** Who knows about this information? **A:** Linda, David, Sally
**Q:** Does Sally know about this information? **A:** Yes

Figure 1: An example question set in 👻 FANToM.

in interactions – i.e., conversations. As conversations present interactions in their raw form, they are much less susceptible to reporting biases, and are more aligned with real-world scenarios requiring ToM reasoning. FANToM consists of 10K questions covering 256 multiparty conversations around a certain topic while characters enter and leave the discussion, leading to distinct mental states between characters due to information asymmetry.

The goal of FANToM is to effectively measure how well models can track the belief of multiple

characters in conversations where some information may be *inaccessible* to some participants. For example, in Figure 1, Kailey briefly steps away from the conversation to get a cup of coffee, while the others continue discussing Linda's new dog. The information exchanged during Kailey's absence remains unknown to Kailey, and only the information shared after Kailey's return becomes accessible. We convert factual question-answer pairs to obtain multiple challenging questions about characters' beliefs concerning the inaccessible information. Our aim is to design questions at different levels that evaluate a model's capability for a coherent understanding of others' mental states. In doing so, we are particularly interested in identifying instances of *illusory ToM*, which we define as situations where a model may answer some questions correctly but fails to answer others that require the same type of ToM reasoning.

The analysis of evaluation results on FANToM reveals several interesting findings (§4): (1) First, existing neural models score significantly lower than humans on individual questions and on the full set of questions by more than 70% on average. (2) While chain-of-thought reasoning (CoT) does improve performance in most models, it does not substantially bridge the gap with human performance. (3) Although our benchmark is not meant for training, we observe that fine-tuning can help models achieve scores higher than human performance on individual question types. However, when it comes to metrics that require coherent responses across multiple question types, the fine-tuned model still significantly underperforms compared to humans. (4) Additionally, we find that models exhibit different error types depending on the format of questions, despite all questions requiring the same underlying reasoning. (5) Moreover, our results indicate that CoT has a selective impact on performance, showing improvement only in specific scenarios.

To the best of our knowledge, FANToM is the first benchmark to introduce conversation-based ToM evaluation for language-based models. Our benchmark design and experiment results yield important insights into the debate around ToM (Whang, 2023) and the development of artificial general intelligence (Metz, 2023) in LLMs. We release our benchmark to spark further discussions on evaluating the ToM capabilities of LLMs.

## 2 Design Considerations for 👻 FANToM

We go over the important design choices that we made when constructing FANToM. Our goal is to incorporate (1) social interactions that necessitate natural theory of mind (ToM) reasoning (§2.1), (2) essential theoretical prerequisites for validating ToM from psychology (§2.2), and (3) empirical findings that must be taken into account when evaluating large language models (§2.3).

### 2.1 Grounding in Social Interactions

To capture the interactive aspect of ToM, we ground our task in natural social interactions – i.e., conversations. By doing so, we gain two key benefits: (1) minimizing reporting bias (Gordon and Van Durme, 2013) and (2) aligning with real-world scenarios.

Since narratives are condensed descriptions of interactions, the process of deciding what to include or exclude can introduce reporting bias, resulting in artifacts that models exploit. For instance, including "*Carlos did not see this, so he does not know currently where the apple is.*" in a narrative for ToM evaluation provides a significant clue about the other's mental state. However, such explicit hints are rarely present in real-world interactions.

Conversations, on the other hand, present interactions in their raw form, without those explicit hints about others' mental states. During conversations, we reason through the intermediate steps from scratch, thereby grounding the benchmark in conversations enables a more realistic and unbiased assessment of ToM.

### 2.2 Meeting Theoretic Requirements

We follow the *two* important criteria outlined by Quesque and Rossetti (2020) that must be met when designing a task to validate ToM: "*non-merging*" and "*mentalizing*".

(1) "*Non-merging*": Evaluation should require the respondent to maintain a distinction between the others' mental state and its own. For example, suppose someone is asked about the other's belief regarding the location of the TV remote controller, and both are believing it to be on the sofa. If the respondent answers that the other believes it is on the sofa, it becomes unclear whether the response is based on the respondent's own belief or the other's (i.e., *merging mental states*). Such *merging* scenario is unsuitable for validating ToM.

Since machines lack emotions or intentions (Gros et al., 2022), we exploit *information asymme-*

*try* when constructing our benchmark to simulate the non-merging mental state scenarios. We design multiparty conversations where specific information is inaccessible to certain characters. While machines do not possess their *own point of view*, they act as omniscient observers during our evaluation since we provide the entire conversation as input. As a result, the mental states of the model and the character can be regarded as distinct with respect to that information.

(2) "*Mentalizing*": Lower-level processes should not be accounted for successful performance of ToM tasks. If a simpler process can explain a phenomenon, it should always be preferred over a more complex one when interpreting the results. For instance, recognizing joy by observing laughter is more of a visual discrimination than reasoning mental representations.

If the correct answer for a ToM task has a high degree of word correlation with a salient part of the given input, it becomes difficult to determine whether the model is accurately ascribing the other's mental state or simply following a shortcut pattern matching (i.e., the lower-level process). Therefore, such cases should be discouraged when evaluating ToM in neural language models. In FANTOM, we create false answers that have high word correlation with the input to verify whether the models can overcome the shortcut pattern matching when reasoning mental states.

## 2.3 Seeking Comprehensive Evaluation

Since the performance of LLMs varies significantly based on given prompts (Webson and Pavlick, 2022), we adopt a series of reiterative questions at various levels for the same input context, including free-form response questions, multiple-choice questions, and straightforward yes or no questions. The inclusion of free-form response questions is important as it aligns with the common usage of LLMs in contrast to multiple-choice questions that are prevalent in existing benchmarks (Sakaguchi et al., 2021; Hendrycks et al., 2021). Although their formats are different, all questions in FANTOM fundamentally aim to ascertain the same underlying reasoning: "*who is aware of the information?*" As a result, FANTOM enables us to identify *illusory ToM* instances wherein models deliver accurate responses for one format but struggles to do so for another format.

## 3 👻 FANTOM Overview

Following the success of previous works (Kim et al., 2022; Chen et al., 2023), we automatically construct full conversations using the large language model (LLM) InstructGPT davinci-003 (Ouyang et al., 2022). We also generate theory of mind (ToM) question-answer pairs related to the conversation participants' beliefs using a specially designed pipeline. In preliminary explorations, we find off-the-shelf LLMs struggle with directly generating ToM question-answer pairs for a given conversation. Our pipeline consists of three steps: (1) generate conversations with information asymmetry (§3.1), (2) generate fact question-answer (QA) pairs (§3.2), and (3) construct ToM (e.g., belief) QA pairs from the fact QA pairs (§3.3). We use different evaluation methods for each question types (§3.4), and validate the final dataset (§3.5).

## 3.1 Information-Asymmetric Conversations

FANTOM consists of small talk conversations involving multiple characters, with each conversation centered around a topic (e.g., pets, risk-taking, personal growth). Each topic has several subtopics, e.g. the topic "*pets*" may include subtopics "*breed*" and "*special moves*". Initially, the conversation begins with two or three characters. As the conversation progresses, characters join and leave the discussion and the conversation's subtopic changes over time. Conversations include explicit indications of leaving and joining, such as utterances like "*Hey guys, I'll go grab a coffee.*" or "*Hey, I'm back, what are you guys discussing now?*" shown in Figure 1. During the absence of a character, the conversation continues and information is shared among the remaining participants, creating a natural information asymmetry that reflects real-life interactions. After a series of utterances, the character who was absent (re)joins the conversation, unaware of the information that was previously shared with other participants. More details are in Appendix A.1.

Many existing ToM tasks involve some form of asymmetry between characters (Braüner et al., 2020). For example, in the Sally-Anne task, Sally does not know that Anne relocated the object, while the observer is aware of the action. In the Smarties task, the character in the story does not know the label changed, whereas the observer is fully aware of this situation. This inherent asymmetry ensures two distinct mental states (i.e., the non-merging criterion; §2.2) to be present during the experiments.

## 3.2 Factual Question-Answer (QA) Pairs

The conversations in FANTOM include factual question-answer pairs (FACTQ) about the inaccessible information—i.e., the information that a specific character is unaware of. An example question would be "*What is the breed of Linda's dog?*" in Figure 1. More details are in Appendix A.2.

There are two distinct types of answers for each FACTQ: (1) FULL FACT A and (2) LIMITED FACT A. The FULL FACT A incorporates the full information in the preceding conversation where the character *PersonX* was absent. On the other hand, LIMITED FACT A relies only on the conversation in which PersonX participated. The former answer is based on information that PersonX does not have access to, while the latter answer only takes into account the accessible information for PersonX. For cases where no information was shared regarding the FACTQ, the LIMITED FACT A indicates that no information has been provided. Examples of these answers are in Figure 1. "*Linda has a golden retriever.*" is based on the preceding conversation where Kailey was absent, while "*There is no information on the breed of Linda's dog*" is based on the conversation Kailey participated.

## 3.3 ToM QAs based on Fact QAs

For each FACTQ, we build six types of ToM QA. Construction details can be found in Appendix A.3.

**(1) BELIEFQ[Dist.] and (2) BELIEFQ[Choice]:** These questions are created by rephrasing the FACTQ to ask beliefs of characters in the conversation. We are particularly interested in PersonX's belief about the inaccessible information from the previous conversation, in which PersonX did not participate. For example, the FACTQ of "*What is the breed of Linda's dog?*" in Figure 1 is converted to "*What breed would Kailey think Linda's dog is?*" The BELIEFQ[Dist.] requires free-form response, while BELIEFQ[Choice] provides multiple-choice options for the same question.

The options for BELIEFQ are created by rephrasing the FULL FACT A and LIMITED FACT A. For example, the "*Linda has a golden retriever.*" in Figure 1 is converted to "*Kailey believes Linda has a golden retriever.*" Since the FULL FACT A reflects information that is not accessible to PersonX and the LIMITED FACT A incorporates only the information accessible to PersonX, we label the converted FULL FACT A and LIMITED FACT A as

"OMNISCIENT-VIEW BELIEF A" and "PERSONX-CENTRIC BELIEF A", respectively.

**(3) ANSWERABILITY Q[List]:** Given the FACTQ, we ask models "*List all the characters who know the correct answer to this question*". In essence, we are interested in whether the model can identify who among the participants can correctly answer the FACTQ. This is a meta-question that necessitates two-step reasoning: first determining the answer itself, and second, identifying the characters who have access to this knowledge.

**(4) INFOACCESS Q[List]:** Here, we provide the FULL FACT A with the FACTQ and ask the model "*List all the characters who know this information*". Essentially, this question aims to identify the individuals who have knowledge or access to this information. Since the information is explicitly provided to the model, only the second reasoning step of the ANSWERABILITY Q[List] is required.

**(5) ANSWERABILITY Q[Y/N] and (6) INFOACCESS Q[Y/N]:** We ask models to determine, through a simple binary response (*yes* or *no*), whether each character is capable of answering the question or knows the information. For example, we ask models "*Does David know the correct answer to this question?*" and "*Does Sally know about this information?*" (Figure 1).

## 3.4 Evaluation

Each question is provided to the model along with the conversation as input. This makes the model an omniscient observer, having access to all information shared in the conversation. On the other hand, PersonX was absent for a while, thereby an information asymmetry naturally arises between the model and PersonX. Responses that include inaccessible information for PersonX indicate a lack of ToM in the model.

**Input context types** 👻 FANTOM comprises two types of input conversations: short and full. In the case of short input, the model is provided with the conversation that only includes the part where the specific speaker left and (re)joined, while excluding the other earlier and later parts of the conversation. On the other hand, a full conversation encompasses the entire discussion on the main topic, including all subtopics. As a result, this is significantly longer than the short input.

**BELIEFQ[DIST.]**    When given a belief question regarding PersonX, the model should generate a response that incorporates only the information accessible to PersonX. We use cosine similarity to measure the distance between SentenceBERT (Reimers and Gurevych, 2019) embeddings of each option and response. A correct response should always be closer to the PERSONX-CENTRIC BELIEF A than the OMNISCIENT-VIEW BELIEF A.

To accurately assess the performance of the response, we also calculate the token F1 score for responses that are considered correct based on the distance metric, following the convention of various QA tasks (Rajpurkar et al., 2016, 2018). When comparing distances in the embedding space, nonsensical responses (e.g., repetition of character names) can be deceptively closer to PERSONX-CENTRIC BELIEF A, resulting in misleading accuracy. Therefore, models must score high on both the distance and F1 metrics for the BELIEFQ[DIST.].

**BELIEFQ[CHOICE]**    The model should choose between the OMNISCIENT-VIEW BELIEF A and the PERSONX-CENTRIC BELIEF A. The correct answer is the PERSONX-CENTRIC BELIEF A.

**ANSWERABILITY Q[LIST] and INFOACCESS Q[LIST]**    A correct response must include all characters who have access to the answer or information while excluding all characters who do not. No partial marks are assigned.

**ANSWERABILITY Q[Y/N] and INFOACCESS Q[Y/N]**    The model should respond with "*yes*" or "*true*" for all characters who have access to the answer or information, and with "*no*" or "*false*" for all characters who do not. More details are in Appendix A.4.

### 3.5   Dataset Validation & Statistics

**Validation**    To ensure the quality of our benchmark, we go through a manual validation process for all conversations and question-answer pairs using Amazon Mechanical Turk (MTurk). We conduct validation on the entire conversations in our dataset using 32 annotators who passed a qualification test for assessing conversation coherence. We ask workers to flag conversations that are incoherent or unsafe (e.g., unethical, biased, harmful, dangerous, or offensive). Each conversation is validated by three workers. While 10 conversations received votes for incoherence, none achieved a majority vote indicating they were incoherent. We

refine all 10 conversations. As for safety, no conversations were voted as being unsafe. We also request workers to verify the answers provided for BELIEFQ[CHOICE]s. We remove all question sets that were marked as erroneous by the worker (~8.6%).

**Statistics** 👻 FANToM is composed of 256 conversations with 1,415 BELIEFQ[DIST.]s and BELIEFQ[CHOICE]s, 703 FACTQs, ANSWERABILITY Q[LIST]s, and INFOACCESS Q[LIST]s, respectively. Additionally, there are 2,689 ANSWERABILITY Q[Y/N]s and INFOACCESS Q[Y/N]s. Given that the ANSWERABILITY Q[Y/N]s and INFOACCESS Q[Y/N]s iterate over all characters present in the conversations, they have the highest count among all the question types.

The average number of turns in the input context is 13.8 (short conversation), and the average number of words in each turn is 21.9. For reference, the corresponding statistics for ToMi (Le et al., 2019) are 4.9 and 4.2, respectively. More statistics can be found in Appendix A.5.

## 4   Experiments

**Baseline Models**    We test a total of thirteen recent instruction-tuned neural language models: GPT-4 (gpt-4-0613 and gpt-4-0314; OpenAI, 2023), ChatGPT (gpt-3.5-turbo-0613; OpenAI, 2022), InstructGPT (davinci-003 and curie-001; Ouyang et al., 2022), Flan-T5-XL and Flan-T5-XXL (Chung et al., 2022), Flan-UL2 (Tay et al., 2023), Falcon Instruct (7B and 40B; Almazrouei et al., 2023), Mistral Instruct 7B (Jiang et al., 2023), Zephyr 7B (HuggingFace, 2023), and Llama-2 Chat 70B (Touvron et al., 2023). Descriptions for each model are in Appendix B.

Although our benchmark is not meant for training, we also fine-tune Flan-T5-XL (Chung et al., 2022) by randomly splitting FANToM according to the conversation's main topics. We then test the model on unseen conversation topics. More details can be found in Appendix B.

**Human Performance**    We also measure human performance by asking graduate students in computer science. We ask BELIEFQ[CHOICE], ANSWERABILITY Q[LIST], and INFOACCESS Q[LIST], given a conversation. As it is redundant to ask human testees binary questions when they have already been asked ANSWERABILITY Q[LIST] and INFOACCESS Q[LIST], we do not ask ANSWERABILITY Q[Y/N]

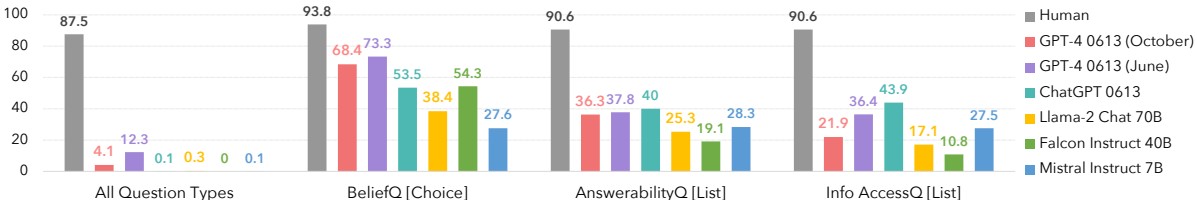

Figure 2: Results of BELIEFQ[CHOICE], ANSWERABILITY Q[LIST] and INFOACCESS Q[LIST], given the short conversation context. Full results with all models, input types, and metrics are in Table 9.

and INFOACCESS Q[Y/N]. To ensure a fair comparison with the models, we give the same instructions to humans and no other tutorials, examples, or extra instructions were given. Student volunteers solved 32 sets in total.

**Metrics** We report accuracy for BELIEFQ[DIST.], BELIEFQ[CHOICE], ANSWERABILITY Q[LIST], and INFOACCESS Q[LIST]. The weighted F1 scores are reported for ANSWERABILITY Q[Y/N] and IN-FOACCESS Q[Y/N]. We additionally report the "*All*" score for the ANSWERABILITY Q and IN-FOACCESS Q requiring models to be correct on both list-type and binary-type questions. For BELIEFQ[DIST.] and FACTQ, we also report the token F1 scores to measure the word overlap between the answer and model's free-form response.

Moreover, we report the ALL* score which requires the models to answer all six ToM question types (§3.3) in the set correctly for the same information piece in the conversation. This metric aims to measure how well the models show consistent understanding across different types of questions. To compare with human performance, we also report the ALL score, which only excludes the BELIEFQ[DIST.] from the ALL* score.

### 4.1 Results

All the models exhibit scores that are significantly worse than human performance. Table 9 shows the full results of state-of-the-art large language models (LLMs) on 👻 FANTOM. We break down the table and highlight each discussion point below.

**Illusory Theory of Mind** Figure 2 shows the results of a few selected models. We find models perform significantly better on BELIEFQ[CHOICE] compared to ANSWERABILITY Q[LIST] and IN-FOACCESS Q[LIST]. Despite the ANSWERABIL-ITY Q[LIST] and INFOACCESS Q[LIST] being prerequisites for solving BELIEFQ[CHOICE], they are much more challenging for models. Furthermore,

| Model | All Question Types | All AnswerabilityQs [List + Y/N] | All InfoAccessQs [List + Y/N] |
|---|---|---|---|
| Human | 87.5 | 90.6 | 90.6 |
| Mistral Instruct + CoT | 0.1 | 2.4 | 9.1 |
| Falcon Instruct + CoT | 0.0 | 1.7 | 2.3 |
| Llama-2 Chat + CoT | 0.4 | 6.0 | 7.8 |
| ChatGPT 0613 + CoT | 3.7 | 20.7 | 17.1 |
| GPT-4 0613 + CoT (Jun) | **26.6** | **40.2** | **57.7** |
| GPT-4 0613 + CoT (Oct) | 14.8 | 31.4 | 41.1 |
| Flan-T5 XL + FT | 53.7 | 55.9 | 54.4 |

Table 1: Results of models with zero-shot chain-of-thought (CoT) and fine-tuning (FT) for the short conversation context. Full results with all models, input types, and metrics are in Table 9.

models' performance sharply drops when evaluated for coherent reasoning across multiple question types with the same underlying theory of mind (ToM) reasoning (i.e., *All Question Types*). These findings suggest that some instances of successful LLM ToM reasoning in FANTOM should be interpreted as illusory.

**Chain-of-thought and Fine-tuning** Table 1 summarizes the results when we apply zero-shot chain-of-thought (CoT) reasoning or fine-tuning to models. For CoT, we follow Kojima et al. (2022) and use the prompt "*let's think step by step*". We observe an improvement in scores with CoT applied. However, there are still significant score gaps compared to human performance.

We also find fine-tuned Flan-T5 XL still falls short of human performance in metrics that demand consistent accuracy across multiple questions—i.e., the *All* scores.[3] Although our benchmark is not intended for training purposes, developing models with a coherent ToM reasoning remains challenging, even with explicit training on the data.

**Comprehending Facts vs. Distinct Beliefs** Figure 3 shows the token F1 scores for FACTQ and ac-

---

[3]We find fine-tuning achieves scores comparable with human performance on individual question types (see Table 9).

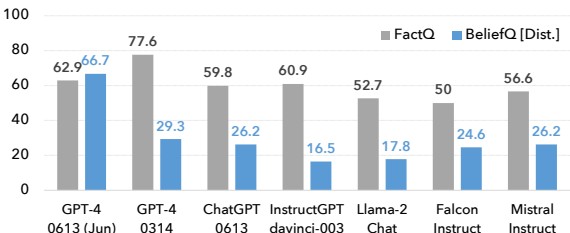

Figure 3: Results of FACTQ and BELIEFQ[DIST.] for models given the short conversation context. Full results with all models, input types, and metrics are in Table 9.

| Model | AnswerabilityQs [Y/N] | InfoAccessQs [Y/N] |
|---|---|---|
| Mistral Instruct 7B | 61.5 | 70.4 |
| Falcon Instruct 40B | 59.4 | 72.2 |
| Llama-2 Chat 70B | 61.4 | 80.4 |
| InstructGPT davinci-003 | 67.0 | 78.4 |
| ChatGPT 0613 | 64.2 | 73.2 |
| GPT-4 0314 | 64.0 | 76.3 |
| GPT-4 0613 (June) | **85.9** | 90.3 |
| GPT-4 0613 (October) | 75.7 | **91.5** |

Table 2: Results of ANSWERABILITY Q[Y/N] and IN-FOACCESS Q[Y/N] when given the short conversation context. Full results with all models, input types, and metrics are in Table 9.

curacy for BELIEFQ[DIST.]. The token F1 scores for FACTQ can be seen as a measure of a model's basic comprehension capability for interactions. Scoring high in FACTQ indicates the model is good at identifying the most relevant information piece to answering the question. Despite its small size, Mistral Instruct 7B shows the strongest performance among the open-source models.

On the other hand, BELIEFQ[DIST.] aims to measure a model's understanding of individual characters' perspective of a particular information—i.e., belief. To meet the *mentalizing* criterion (see §2.2), we deliberately design the incorrect answers in BELIEFQ[DIST.] to have greater word overlap with the context than correct answers. Also, BELIEFQ[DIST.] are rephrased questions inquiring about PersonX's belief for the facts in FACTQ, thereby the two question types share significant word overlap. However, the same information that was used to answer FACTQ should not be included in the response for BELIEFQ[DIST.] on PersonX as it is from the conversation that PersonX missed. As a result, certain models with higher token F1 scores for FACTQ have lower scores for BELIEFQ[DIST.] compared to models that perform worse on FACTQ (e.g., InstructGPT davinci-003 vs. Llama-2 Chat and Mistral Instruct). This suggests the models lack the ability to comprehend distinct perspectives of individual characters, leading them to reproduce similar responses to FACTQ for BELIEFQ[DIST.].

**Free-Response vs. Choice** We observe a pattern where models score significantly worse in free-response questions than choice questions (BELIEFQ[DIST.] vs. BELIEFQ[CHOICE]; Figure 3 and 2).[4] However, many of them still achieve scores either below or around 50, which is the random baseline for those binary choice questions.

**Reasoning Complexity** Table 2 compares models' performance between ANSWERABILITY Q[Y/N] and INFOACCESS Q[Y/N]. As ANSWERABILITY Qs require an additional step of reasoning compared to INFOACCESS Qs, models consistently perform worse on ANSWERABILITY Q[Y/N] compared to INFOACCESS Q[Y/N]. However, this pattern is not consistent across models for ANSWERABILITY Q[LIST] and INFOACCESS Q[LIST] (see Figure 2). This may be because models significantly struggle with ANSWERABILITY Q[LIST] and INFOACCESS Q[LIST], potentially resulting in the absence of meaningful performance patterns.

**Short vs. Full Conversations** When a model is provided with the full conversation (Table 9, bottom), its performance noticeably decreases compared to when it is given only the relevant parts of the conversation (Table 9, top). The decrease can be attributed to the model's need to identify the relevant information within the full conversation, whereas it does not have to do so for the short conversations. This indicates theory of mind reasoning becomes even more challenging for models when it needs to be combined with different types of reasoning (e.g., search).

### 4.2 In-depth Analysis

**What types of errors do models make?** Figure 4 and 5 summarize the error types of ANSWERABILITY Q and INFOACCESS Q for each model with and without chain-of-thought (CoT) reasoning. For list-type questions, models make more errors by including characters who are unaware of the information in the responses, rather than excluding characters who are aware. Interestingly, when CoT is applied, the error of including unaware characters decreases, whereas the error of excluding characters who are aware increases for most models.

---

[4]This pattern is consistent for ANSWERABILITY Q[LIST] and ANSWERABILITY Q[Y/N], as well as for INFOACCESS Q[LIST] and INFOACCESS Q[Y/N] (see Table 9).

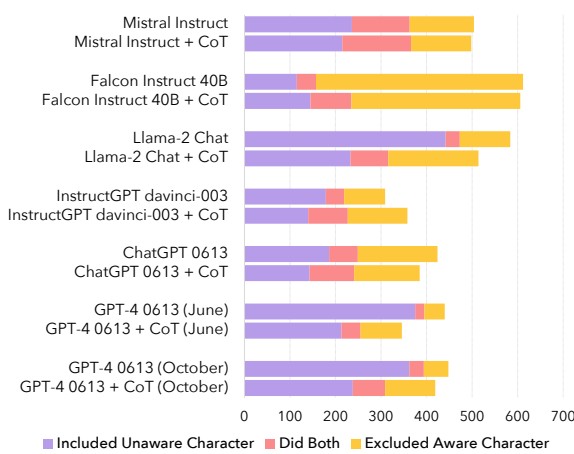

Figure 4: Analysis of model errors for ANSWERABIL-ITY Q[LIST] and INFOACCESS Q[LIST].

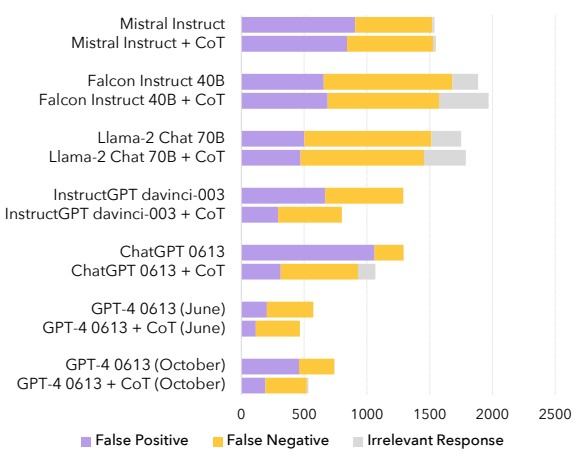

Figure 5: Analysis of model errors for ANSWERABIL-ITY Q[Y/N] and INFOACCESS Q[Y/N].

In the case of binary questions, false positives and false negatives correspond to including characters who are unaware and excluding characters who are aware in the response for list-type questions, respectively. If the model fails to generate a yes or no response, we mark it as irrelevant. Models tend to exhibit false negative responses more frequently for binary questions compared to list-type questions. Similarly, CoT primarily helps the model in reducing the false positive error rates, but the reduction in false negative error rates is not consistent across models. This suggests that CoT selectively improves reasoning specifically for determining characters who are unaware of the information, rather than characters who are aware.

**How accurate and consistent are models' answers for a given character?** For accuracy, we report the ALL FOR EACH CHARACTER score which is determined by whether the models are able

| Model | ALL FOR EACH CHARACTER | Answer Consistency |
|---|---|---|
| Mistral Instruct | 27.8 | 45.1 |
| Mistral Instruct + CoT | 26.9 | 41.9 |
| Falcon Instruct 40B | 10.7 | 19.1 |
| Falcon Instruct 40B + CoT | 16.9 | 27.4 |
| Llama-2 Chat 70B | 27.1 | 43.3 |
| Llama-2 Chat 70B + CoT | 15.2 | 24.3 |
| InstructGPT davinci-003 | 33.1 | 55.2 |
| InstructGPT davinci-003 + CoT | 35.2 | 58.4 |
| ChatGPT 0613 | 35.0 | 51.6 |
| ChatGPT 0613 + CoT | 31.5 | 44.9 |
| GPT-4 0613 (June) | 53.2 | 66.8 |
| GPT-4 0613 + CoT (June) | 59.2 | 73.4 |
| GPT-4 0613 (October) | 48.7 | 62.2 |
| GPT-4 0613 + CoT (October) | 51.2 | 66.9 |

Table 3: The accuracy and consistency (%) of the models' responses for each character within the given conversation context.

| Model | First-Order | Second-Order | | |
|---|---|---|---|---|
| | | Overall | Cyclic | Acyclic |
| Mistral Instruct | 22.0 | 30.3 | 35.5 | 25.1 |
| Mistral Instruct + CoT | 27.0 | 39.5 | 40.8 | 38.3 |
| Falcon Instruct 40B | 39.3 | 41.1 | 42.6 | 39.6 |
| Falcon Instruct 40B + CoT | 67.9 | 76.3 | 75.5 | 77.2 |
| Llama-2 Chat | 15.0 | 20.5 | 21.0 | 20.1 |
| Llama-2 Chat + CoT | 29.5 | 33.5 | 32.7 | 34.3 |
| InstructGPT davinci-003 | 15.4 | 19.4 | 23.2 | 15.6 |
| InstructGPT davinci-003 + CoT | 23.9 | 20.5 | 23.7 | 17.3 |
| ChatGPT 0613 | 21.2 | 31.1 | 31.2 | 30.9 |
| ChatGPT 0613 + CoT | 47.8 | 42.6 | 44.9 | 40.4 |
| GPT-4 0613 (June) | 63.8 | 66.7 | 66.3 | 67.1 |
| GPT-4 0613 + CoT (June) | 65.9 | 67.6 | 69.1 | 66.0 |
| GPT-4 0613 (October) | 49.1 | 63.0 | 63.1 | 62.9 |
| GPT-4 0613 + CoT (October) | 45.8 | 64.0 | 62.6 | 65.4 |

Table 4: BELIEFQ results for first and second order ToM beliefs.

to answer *all* six types of ToM questions correctly regarding the specific character. For consistency, we measure the ratio of consistent model responses across ANSWERABILITY Q and INFOACCESS Q for each character. Table 3 shows the accuracy and consistency of the models' responses for each character within the given conversation context. Overall, we observe a pattern where models that score low in accuracy also show low consistency. While CoT generally improves model performance (see Table 9), we find that it does not always lead to improved accuracy and consistency. The decrease in ALL FOR EACH CHARACTER score when CoT is applied suggests that CoT has a selective impact on different question types.

**Are there differences in performance in terms of the order of ToM beliefs?** Table 4 presents the results of BELIEFQ with respect to different

orders of ToM beliefs. Similar to Le et al. (2019), models perform better on the second-order belief questions than those with first-order beliefs. To further investigate the performance on second-order belief questions, we analyze the results based on the cyclic and acyclic patterns in them. The cyclic second-order belief questions inquire about Character 1's belief regarding Character 2's belief about Character 1 (e.g., *What does Linda think about Kailey's belief on the breed of Linda's dog?*); while the acyclic second-order questions focus on Character 1's belief about Character 2's belief regarding Character 3 (e.g., *What does David think about Kailey's belief on the breed of Linda's dog?*). Models show better performance on the cyclic questions than acyclic ones, which include more characters to track. However, when CoT is applied, the increase in score for acyclic questions is greater than that of cyclic ones, suggesting CoT helps multi-tracking.

## 5 Related Work

**Existing Theory of Mind Benchmarks** Many theory of mind (ToM) benchmarks, inspired by the false belief test from psychology (Wimmer and Perner, 1983), evaluate models on reasoning beliefs about object locations with narratives (Grant et al., 2017; Nematzadeh et al., 2018; Le et al., 2019). Other works such as Shapira et al. (2023b) build benchmarks based on the Faux Pas Test (Baron-Cohen et al., 1999). Also, ToM-related benchmarks focus on reasoning emotions and mental states in narratives (Rashkin et al., 2018; Sap et al., 2019).

**Theory of Mind in Large Language Models** Although qualitative assessments might imply a degree of ToM in large language models (LLMs; Whang, 2023), more comprehensive quantitative investigations reveal that they have yet to achieve human-level ToM across various benchmarks (Sap et al., 2022; Shapira et al., 2023a). LLMs struggle to reason ToM robustly (Ullman, 2023), though their performance can be improved through few-shot samples and chain-of-thought prompting (Sap et al., 2022; Moghaddam and Honey, 2023) as well as specific inference methods (Sclar et al., 2023).

## 6 Conclusion & Discussion

We introduced 👻 FANTOM, a new benchmark for stress-testing theory of mind (ToM) capabilities of neural language models in conversations via question answering. Our benchmark is built upon essential theoretical requisites and empirical considerations required for validating ToM in large language models (LLMs). The conversations in our benchmark involve information asymmetry, with characters joining and leaving the discussion while it continues, to simulate distinct mental states. To identify illusory ToM, we crafted multiple types of challenging belief questions regarding the conversation participants' mental states by converting factual questions. Our evaluation results show that coherent ToM reasoning is challenging for current LLMs, performing significantly worse than humans even when using chain-of-thought reasoning or fine-tuning.

Although there has been recent debates around whether current LLMs possess ToM capabilities or not (Whang, 2023), our results indicate that this capacity has not yet emerged in any manner. Previous instances of success on well-known psychology ToM tests may be attributed to exposure during the pretraining phase (Ullman, 2023). Our work highlights the need for novel interaction-oriented benchmarks that introduce scenarios not encountered during training, and also aligning more closely with real-world use cases as LLMs are increasingly being deployed in interactive settings.

Our results also shed light on a broader issue in neural models – the lack of internal consistency (Elazar et al., 2021). We find they often fail to provide consistent answers to questions requiring the same underlying ToM reasoning. To address this concern, future works can explore various directions, such as grounding reasoning in pragmatics (Kim et al., 2020), visual information (Bisk et al., 2020), or belief graphs (Sclar et al., 2023).

Another issue that our work touches upon is the reporting biases inherent in language models. We observed that models often exhibit biases in their responses, showing a tendency to overly rely on the information they are conditioned on, such as preferring answers that have high overlap with the context (Sugawara et al., 2018). However, to achieve successful ToM reasoning, it is crucial to distinguish between accessible and inaccessible information for a particular agent, rather than blindly using all information available to the model. One potential approach to mitigate this is to combine pretraining with interactive learning (Sap et al., 2022).

In the spirit of encouraging future research in this direction, we make our benchmark publicly available at `https://hyunw.kim/fantom`.

## 7 Limitations

Although FANTOM is the first benchmark, to the best of our knowledge, to cover theory of mind (ToM) reasoning in conversational interactions, it is currently limited to small talks on specific topics. Additionally, our benchmark only considers only a single type of relationship between conversation participants, where they do not have prior knowledge of each other. However, social reasoning can become much more dynamic when variables such as relationships (e.g., family, friends, co-workers) are introduced. ToM is essential in all conversational interactions, hence we strongly encourage future works to evaluate ToM in a wider range of diverse conversation scenarios.

Our evaluation solely focuses on language-based models. However, it is important to note that ToM extends beyond a single modality (Piaget, 1956; Wu and Keysar, 2007). For instance, the well-known Sally-Anne test (Wimmer and Perner, 1983; Baron-Cohen et al., 1985) is typically conducted as a face-to-face experiment, where visual cues affect the performance of the participants. Therefore, interesting future work will involve examining the capabilities of multi-modal models in relation to ToM reasoning.

Lastly, as we generate full conversations with large language models, conversations may contain offensive contents (Weidinger et al., 2021). However, we specifically select casual topics for small talks (e.g., pets, personal growth, traveling) to minimize the likelihood of offensive content generation. Also, we manually validate all conversations in our benchmark with crowdworkers from Amazon Mechanical Turk.

## 8 Societal and Ethical Considerations

We acknowledge that the term "*theory of mind*" (ToM) may evoke anthropomorphic connotations regarding AI models. However, we emphasize that the purpose of our work is not to promote anthropomorphism of AI models. Rather, our focus lies in exploring the limitations of existing language models in social reasoning. While the concept of ToM attempts to capture the ability to attribute mental states to oneself and others (Premack and Woodruff, 1978), it is important to clarify that AI models do not possess subjective consciousness or true understanding of intentions, beliefs, or desires. Our experiment results also demonstrate that current large language models do not exhibit any coherent ToM reasoning; instead, they primarily rely on word correlations.

## Acknowledgement

We thank the participants who contributed to the human performance measurement. We also appreciate our colleagues on the Beaker Team at the Allen Institute for AI for helping with the compute infrastructure. This work was supported in part by DARPA MCS program through NIWC Pacific (N66001-19-2-4031). Hyunwoo Kim and Gunhee Kim are supported by the Institute of Information & communications Technology Planning & Evaluation (IITP) grant funded by the Korea government (MSIT) (No.2019-0-01082, SW StarLab; and No.2022-0-00156, Fundamental research on continual meta-learning for quality enhancement of casual videos and their 3D metaverse transformation). Lastly, we also thank OpenAI, as well as Google Cloud Compute.

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

# A 👻 FANToM Construction

Full examples of question sets in FANToM can be found in Table 5 and Table 6.

## A.1 Generating Conversations with Information Asymmetry

**Information-asymmetric conversations** To create the conversations in our benchmark, we use a predefined set of subtopics for each main topic and employ templates to generate scripts. For example, for the topic "*pets*" subtopics may include "*breed*", "*special moves*", and "*favorite food*". Following Kim et al. (2022), we use specific speaker prefixes with English names sampled from the Top-1K names in the US SSN database for more natural conversations. We append each utterance with speaker prefixes. We randomly shuffle the subtopics for each topic and generate conversations for each subtopic. We generate the first conversation with the following prompt: "`{Character 1}, {Character 2}, ... {Character n} met for the first time at this social event. They are having a conversation on their {topic}. They now discuss {subtopic}.\n{Character 1}:`" The initial conversation starts with two or three characters and there can be up to five characters who are participating in the conversation at the same time.

Then, for each subtopic, we randomly select characters to join or leave the conversation. We use the following prompt when a character is selected to leave: "`Now, {leaving character} leaves the conversation because of the reason '{leaving reason}'. They now discuss {subtopic}. Remember to indicate that {leaving character} is leaving the`

conversation. `{Conversation history} \n{leaving character}:`". We use a predefined list of 64 reasons for leaving the conversation. Table 7 shows all reasons for leaving. We append the previous conversation history to the input prompt to make the conversation continue from the previous one.

We use the following prompt when a character is selected to join: "`Now {joining character} comes back after leaving the conversation because of the reason {leaving reason}. They now discuss {subtopic}. Remember to indicate that {joining character} is joining the conversation. Do not mention the details in the previous conversations. {Conversation history} \n{joining character}:`".

**Extracting the inaccessible information for PersonX** Whenever a character (re)joins the conversation, we extract the inaccessible information by asking GPT-4 (`gpt-4-0314`) what information was shared in the preceding conversation where the character `PersonX` did not participate. We provide the previous conversation and the current one as input to GPT-4 with the prompt "`What information was shared before PersonX joined, but was not mentioned after PersonX joined?`" appended to it. To ease the task, the joining of the character is explicitly denoted by inserting a script between the conversations, as follows: "`Previous conversation\n[PersonX joined the conversation]\nCurrent conversation`". We observe quality improvements for the output generated by GPT-4 with the inclusion of the hint script. The returned result can be viewed as a conversation summary explicitly covering the previous context.

## A.2 Generating Factual QA Pairs

We construct factual question-answer (QA) pairs related to the inaccessible information. First, we generate three non-yes-or-no questions and denote these as "FACTQs" and obtain them by prompting GPT-4, given the inaccessible information text. We obtain "FACTQs" by prompting GPT-4 with the following: "`{inaccessible information}\n\nBased on this, formulate three non-yes-or-no questions that can be answered by this conversation summary.`"

Next, we generate two distinct types of answers for each FACTQ with GPT-4. (1) First, we gener-

...
**Sabrina:** So, what was the most challenging workout experience you ever had?
**Anna:** Definitely when I decided to try out CrossFit. I'm not going to lie, it kicked my butt!
**Sabrina:** Wow, that sounds intense. What kind of exercises did you do?
**Anna:** We did a lot of different things like high intensity interval training and Olympic lifting with barbells and dumbbells.
**Sabrina:** That definitely takes dedication! How did you stay motivated during it?
**Anna:** It was tough but I kept reminding myself why I wanted to get fit in the first place and that helped me stay focused on my goals.
**Gina:** Hey, I'm back! What were you guys talking about?
**Sabrina:** We were just discussing our most challenging workout experiences. What do you think about when it comes to making a good workout playlist?
**Anna:** Music is really important when it comes to getting in the zone while working out. For me, I like upbeat and energetic songs that get me going.
**Gina:** Yeah, something with a high tempo can really help push you during those tough workouts! I also like adding in some of my favorite classic songs that give me extra motivation to keep going.
...

**FACTQ:** In what ways did Anna motivate herself when faced with difficult workouts?

**BELIEFQ:** What does Gina believe are the ways in which Anna motivated herself when faced with difficult workouts?
**OMNISCIENT-VIEW BELIEF A:** Gina believes that Anna motivated herself by reminding herself why she wanted to get fit in the first place and staying focused on her goals.
**PERSONX-CENTRIC BELIEF A:** Gina believes that Anna motivated herself by listening to upbeat and energetic songs during her workouts.

**ANSWERABILITY Q[LIST]:** List all the characters who know the correct answer to this question.
**INFOACCESS Q[LIST]:** List all the characters who know this information.
**Answer:** Sabrina, Anna

**ANSWERABILITY Q[Y/N]:** Does Sabrina know the correct answer to this question?
**INFOACCESS Q[Y/N]:** Does Sabrina know this information?
**Answer:** Yes

Table 5: A sample from 👻 FANToM.

ate an answer denoted as "FULL FACT A", which is based on the preceding conversation where PersonX was absent. This answer incorporates the full information by providing GPT-4 with the previous conversation – i.e., the source of the inaccessible information for PersonX. (2) Second, we generate another answer referred to as "LIMITED FACT A", which relies only on the conversation where PersonX participated. In this case, we give GPT-4 the PersonX-participating conversation along with the FACTQ. We prompt GPT-4 with the following: "{context}\n\nQuestion: {FACTQ}\nAnswer:"

## A.3 Constructing Belief QAs with Factual QAs

**BELIEFQ[DIST.] and BELIEFQ[CHOICE]** We first convert FACTQs into first-order or second-order ToM questions asking about beliefs of characters in the conversation. We are particularly interested in PersonX's belief or knowledge about the inaccessible information from the previous conversation, in which PersonX did not participate. We prompt GPT-4 with the following: "{FACTQ}\n\nConvert this into a theory of mind question asking {character name}'s belief about this."

Next, we convert the FULL FACT As and LIMITED FACT As into answers about beliefs. Since the FULL FACT As reflect information that is not accessible to PersonX and the LIMITED FACT A incorporates only the information accessible to PersonX, we label the converted FULL FACT A and LIMITED FACT A as "OMNISCIENT-VIEW BELIEF A" and "PERSONX-CENTRIC BELIEF A", respectively. For the conversion, we prompt GPT-4 with the following format: "Question: FACTQ \n\nAnswer the question using the following sentence. {FULL FACT A or LIMITED FACT A }\nAnswer:".

## A.4 Evaluation for ANSWERABILITY Q[Y/N] and INFOACCESS Q[Y/N]

We use pattern matching to parse the yes or no answers from model responses. We regard "*yes*", "*knows*", "*does know*", and "*true*" as responses representing "*yes*". Similarly, we regard "*no*", "*does not know*", "*doesn't know*", and "*false*" as responses representing "*no*".

## A.5 Statistics for 👻 FANToM

Table 8 compares the basic statistics of FANToM and ToMi (Le et al., 2019).

## B Experiments

**Human performance evaluation** A total of 11 student volunteers participated in the evaluation. For each question set, we assign a single testee. They solved a total of 32 sets. To ensure a fair comparison, no additional tutorials, examples, or extra instructions were provided beyond what was given to the models.

**Baseline models** The GPT models are proprietary models from OpenAI based on the decoder-only transformer architecture. Flan-T5 and Flan-

UL2 are open-source (i.e., Apache 2.0) models from Google trained on instruction-phrased datasets. They are based on the encoder-decoder transformer architecture. Falcon Instruct is another open-source (i.e., Apache 2.0) model trained on RedefinedWeb (Penedo et al., 2023) and Baize (Xu et al., 2023). Llama-2 Chat (Touvron et al., 2023) is a fine-tuned 70B large language model, optimized for following user requests in dialogue format. Mistral Instruction (Jiang et al., 2023) is a 7B language model fine-tuned to follow instructions, which is reported to surpass the Llama-2 Chat 13B model. Zephyr (HuggingFace, 2023) is a model based on Mistral, further fine-tuned on UltraChat (Ding et al., 2023) and aligned with UltraFeedback (Cui et al., 2023).

**Results of other models**   Table 9 shows the results for other large language models not included in Figure 2. Given the random baseline score is 50 for BELIEFQ[CHOICE], ANSWERABILITY Q[Y/N], and INFOACCESS Q[Y/N], most of the models show low performance on our benchmark.

**Fine-tuning details**   We fine-tune Flan-T5-XL with `learning rate=2e-5` and `weight decay=0.01`, evaluating per epoch and using early stopping with patience 1 (batch size = 3 for Flan-T5-XL). We observe an increase in validation loss after the first epoch. We also add special tokens before and after the completions to prevent the model from over-generating, which we find in early experiments. We also fine-tune text-curie-001 (Ouyang et al., 2022) for two epochs using standard parameters from the OpenAI API.

---

...
**Zachary:** Have you guys thought about how much money you'll need for retirement?
**Hazel:** I'm still trying to figure that out. I know it's important to save, but it can be hard when there are so many other expenses.
**Cory:** Yeah, that's true. It's also good to plan for how you want your lifestyle to look when you retire too. Will you travel or stay close to home?
**Hazel:** That's a tough one! I think a combination of both might work best for me - travel while we're younger and then spend more time closer to family as we get older.
**Zachary:** Sounds like a great plan! What kind of investments have each of you made?
**Cory:** Well, I've started investing in some mutual funds and my 401(k). How about the two of you?
**Hazel:** I've been contributing regularly to my IRA and also investing in index funds. Zachary?
**Zachary:** Same here - mutual funds and an IRA with occasional stock investments as well.
**Hazel:** Sorry guys, but my pet needs attention. I need to go check on him.
**Cory:** Oh, okay. Have a good one!
**Hazel:** See you later.
**Cory:** Well, now that Hazel is gone, we can talk about the psychology of retirement. What are some of the emotional aspects to consider when planning for retirement?
**Zachary:** It can be hard to think about the future and envision what life will be like in retirement without having any prior experience with it. People also tend to underestimate how much money they'll need for a comfortable lifestyle and overestimate their ability to save for it over time.
**Alec:** Hey, I'm back! What were you guys talking about?
**Cory:** We were discussing retirement planning. Zachary was just mentioning how difficult it can be to think about the future and envision what life will be like in retirement without having any prior experience with it.
**Alec:** Yeah, that's true. It's important to save as much as possible now so you have enough money for retirement later. Have you guys started investing yet?
**Zachary:** Yes, we both have investments in mutual funds and IRAs, and Cory has a 401(k). How about you Alec?
**Alec:** I've been saving some money into a savings account and I'm looking into investing in index funds soon. What other advice do you have for someone starting out with their retirement planning?
**Cory:** Do your research on different investment options available to figure out which ones are best for your goals. There are also some tax benefits associated with certain types of investments that could make them more attractive than others depending on your situation.
...

---

**FACTQ:** What kind of funds has Hazel been investing in?

---

**BELIEFQ:** What does Alec believe are the types of funds Hazel has been investing in?
**OMNISCIENT-VIEW BELIEF A:** Alec believes that Hazel has been investing in index funds and contributing regularly to her IRA.
**PERSONX-CENTRIC BELIEF A:** Alec does not know the types of funds Hazel has been investing in.

---

**ANSWERABILITY Q[LIST]:** List all the characters who know the correct answer to this question.
**INFOACCESS Q[LIST]:** List all the characters who know this information.
**Answer:** Hazel, Zachary, Cory

---

**ANSWERABILITY Q[Y/N]:** Does Alec know the correct answer to this question?
**INFOACCESS Q[Y/N]:** Does Alec know this information?
**Answer:** No

---

Table 6: Another sample from 👻 FANTOM.

bathroom break
coffee break
forgot something important
forgot to print some documents
forgot to recieve a package
forgot to return a package
forgot to run errands
forgot to submit documents
have a meeting starting soon that I need to prepare for
have a previous engagement that I need to attend to quickly
have a work-related emergency that requires my immediate attention
have an unexpected visitor at my door
have errands to run
have to attend to someone who just walked in
have to check on something
have to go to the restroom
have to pick up a prescription
have to pick up dry cleaning
have to print or scan documents
have to receive a delivery
have to recharge laptop
have to return a borrowed item
have to take care of a family matter
have to take care of an unexpected task
have unexpected visitor
his/her pet needs attention
his/her family is calling
incoming delivery
must respond to a phone call
need to check on a friend or family member who needs assistance
need to finish a task that's time-sensitive
need to get a phone call
need to get some coffee
need to go to the toilet
need to grab a snack or a drink
need to have a quick chat with someone else
need to make a phone call
need to make a quick trip to the drug store
need to make a quick trip to the grocery store
need to pick up a package
need to receive a parcel
need to recharge cellphone
need to register for an event
need to schedule a haircut or salon appointment
need to schedule another appointment
need to step away for a moment to stretch and clear my mind
need to step out for a moment
need to submit some papers
need to take care of some paperwork or documents
need to take care of some personal matters
need to take care of something related to my health
need to take care of something urgent
need to troubleshoot something
parking meter expiring
remembered something that needs to be taken care of
remembered to receive a package
remembered to submit some papers
remembered to take care of some paperwork or documents
remembered to take care of some personal matters
remembered to take care of something urgent
want to go grab a drink
want to go grab a coffee
want to go take some fresh air
want to go to the bathroom

Table 7: Predefined reasons for characters leaving the conversation.

| Dataset | Total #Questions | Avg. #Questions per Context | Avg. #Turns (Partial) | Avg. #Turns (Full) | Avg. Turn Length |
|---|---|---|---|---|---|
| ToMi | 6K | 6.0 | - | 4.9 | 4.7 |
| 🧑 FANToM | 10K | 12.9 | 13.8 | 24.5 | 21.9 |

Table 8: Statistics of FANToM and ToMi (Le et al., 2019).

| Model | ALL* QUESTION TYPES | ALL QUESTION TYPES | BELIEF QUESTIONS Choice | Dist. | TokenF1 | ANSWERABILITY QUESTIONS All | List | Y/N | INFO ACCESS QUESTIONS All | List | Y/N | FACT QUESTIONS TokenF1 |
|---|---|---|---|---|---|---|---|---|---|---|---|---|
| Human | | | 87.5 | 93.8 | | 90.6 | 90.6 | | 90.6 | 90.6 | | |
| **Short Conversation** | | | | | | | | | | | | |
| Flan-T5-XL | 0.0 | 0.1 | 30.5 | 40.1 | 3.2 | 6.5 | 17.2 | 62.7 | 1.4 | 11.0 | 51.0 | 22.9 |
| Flan-T5-XXL | 0.1 | 0.3 | 27.3 | 42.1 | 2.2 | 2.4 | 15.1 | 54.9 | 1.7 | 10.8 | 50.4 | 22.9 |
| Flan-UL2 | 0.0 | 0.1 | 23.0 | 47.6 | 2.9 | 5.7 | 25.8 | 60.3 | 1.1 | 16.5 | 49.9 | 21.8 |
| Mistral Instruct 7B | 0.0 | 0.1 | 27.6 | 26.2 | 50.8 | 2.4 | 28.3 | 61.5 | 9.1 | 27.5 | 70.4 | 56.6 |
| Zephyr 7B | 0.0 | 0.0 | 58.5 | 42.0 | 41.9 | 0.4 | 12.6 | 60.6 | 2.6 | 35.4 | 61.0 | 55.0 |
| Falcon Instruct 7B | 0.0 | 0.0 | 43.9 | 20.2 | 26.4 | 0.9 | 13.2 | 52.4 | 2.1 | 13.7 | 56.4 | 33.5 |
| Falcon Instruct 40B | 0.0 | 0.0 | 54.3 | 24.6 | 33.6 | 13.4 | 19.1 | 59.4 | 5.8 | 10.8 | 72.2 | 50.0 |
| Llama-2 Chat 70B | 0.0 | 0.3 | 38.4 | 17.8 | 36.0 | 2.4 | 25.3 | 61.4 | 6.5 | 17.1 | 80.4 | 52.7 |
| InstructGPT curie-001 | 0.0 | 0.0 | 21.0 | 14.7 | 42.6 | 0.1 | 7.3 | 54.2 | 0.0 | 3.3 | 58.2 | 47.3 |
| InstructGPT davinci-003 | 0.0 | 0.4 | 17.7 | 16.5 | 44.5 | 9.3 | 56.3 | 67.0 | 16.8 | 33.8 | 78.4 | 60.9 |
| ChatGPT 0613 | 0.0 | 0.1 | 53.5 | 26.2 | 50.8 | 3.1 | 40.0 | 64.2 | 13.3 | 43.9 | 73.2 | 59.8 |
| GPT-4 0314 | 0.4 | 0.6 | 39.0 | 29.3 | 42.8 | 4.4 | 34.7 | 64.0 | 10.1 | 18.2 | 76.3 | 77.6 |
| GPT-4 0613 (June) | 8.2 | 12.3 | 73.3 | 65.3 | 48.2 | 28.6 | 37.8 | 85.9 | 29.0 | 36.4 | 90.3 | 62.9 |
| GPT-4 0613 (October) | 2.4 | 4.1 | 68.4 | 56.1 | 44.6 | 16.9 | 36.3 | 75.7 | 17.9 | 21.9 | 91.5 | 64.9 |
| Flan-T5-XL + CoT | 0.0 | 0.0 | 43.0 | 26.4 | 15.4 | 0.9 | 9.2 | 57.1 | 1.6 | 8.4 | 65.3 | 21.5 |
| Flan-UL2 + CoT | 0.0 | 0.0 | 24.7 | 32.4 | 7.3 | 1.1 | 10.2 | 57.3 | 0.3 | 2.6 | 59.4 | 15.6 |
| Mistral Instruct 7B + CoT | 0.0 | 0.4 | 58.5 | 31.5 | 19.4 | 6.0 | 26.8 | 63.6 | 7.8 | 28.2 | 67.4 | 33.9 |
| Zephyr 7B + CoT | 0.0 | 0.0 | 49.0 | 69.6 | 22.4 | 3.7 | 24.7 | 64.0 | 1.1 | 10.1 | 58.5 | 27.4 |
| Falcon Instruct 7B + CoT | 0.0 | 0.0 | 42.4 | 45.3 | 17.9 | 0.9 | 9.5 | 49.5 | 2.1 | 5.9 | 56.2 | 19.0 |
| Falcon Instruct 40B + CoT | 0.0 | 0.0 | 51.7 | 72.1 | 18.4 | 1.6 | 12.9 | 58.4 | 0.9 | 5.9 | 65.1 | 19.5 |
| Llama-2 Chat 70B + CoT | 0.0 | 0.4 | 58.5 | 31.5 | 19.3 | 6.0 | 26.8 | 63.6 | 7.8 | 28.3 | 67.0 | 33.9 |
| InstructGPT curie-001 + CoT | 0.0 | 0.0 | 12.3 | 16.7 | 36.5 | 0.1 | 7.4 | 58.8 | 0.0 | 2.8 | 58.1 | 38.5 |
| InstructGPT davinci-003 + CoT | 1.3 | 6.2 | 39.8 | 22.2 | 41.6 | 9.3 | 49.4 | 80.4 | 16.8 | 42.9 | 86.6 | 49.9 |
| ChatGPT 0613 + CoT | 2.1 | 3.7 | 58.5 | 45.2 | 44.7 | 20.7 | 45.5 | 76.7 | 17.1 | 36.1 | 79.1 | 53.4 |
| GPT-4 0314 + CoT | 1.0 | 2.8 | 39.0 | 31.2 | 36.8 | 21.9 | 38.1 | 83.7 | 10.7 | 22.5 | 73.2 | 56.2 |
| GPT-4 0613 (June) + CoT | 18.4 | 26.6 | 80.6 | 66.7 | 44.0 | 40.2 | 51.1 | 88.5 | 57.7 | 63.6 | 92.1 | 54.3 |
| GPT-4 0613 (October) + CoT | 6.8 | 14.8 | 74.7 | 55.0 | 40.0 | 31.4 | 40.4 | 86.6 | 41.1 | 46.4 | 91.3 | 52.8 |
| InstructGPT curie-001 + FT | 0.0 | 0.0 | 56.6 | 54.7 | 43.6 | 3.7 | 4.4 | 91.9 | 5.8 | 5.9 | 91.8 | 35.8 |
| Flan-T5-XL + FT | 26.5 | 53.7 | 93.4 | 63.5 | 42.4 | 55.9 | 78.7 | 86.7 | 54.4 | 75.0 | 86.2 | 49.3 |
| **Full Conversation** | | | | | | | | | | | | |
| Flan-T5-XL | 0.0 | 0.0 | 3.8 | 38.2 | 4.7 | 0.0 | 5.8 | 11.1 | 0.3 | 4.4 | 9.9 | 8.7 |
| Flan-T5-XXL | 0.0 | 0.0 | 3.8 | 36.6 | 4.5 | 0.0 | 2.6 | 10.6 | 0.0 | 1.3 | 8.6 | 8.6 |
| Flan-UL2 | 0.0 | 0.0 | 3.8 | 38.9 | 7.5 | 0.9 | 7.1 | 12.9 | 0.0 | 4.6 | 9.1 | 10.6 |
| Mistral Instruct 7B | 0.0 | 0.0 | 25.7 | 25.0 | 51.3 | 1.6 | 25.8 | 55.8 | 5.8 | 19.4 | 64.9 | 53.5 |
| Zephyr 7B | 0.0 | 0.0 | 61.6 | 31.5 | 40.2 | 0.1 | 10.4 | 49.7 | 2.7 | 17.0 | 48.9 | 41.8 |
| Falcon Instruct 7B | 0.0 | 0.0 | 16.8 | 34.8 | 7.0 | 0.1 | 2.8 | 48.1 | 0.1 | 1.4 | 58.3 | 10.7 |
| Falcon Instruct 40B | 0.0 | 0.0 | 13.3 | 56.4 | 16.2 | 0.5 | 16.7 | 58.3 | 0.7 | 18.9 | 58.3 | 23.3 |
| Llama-2 Chat 70B | 0.0 | 0.0 | 49.0 | 37.1 | 27.8 | 1.9 | 16.4 | 52.8 | 2.2 | 11.1 | 69.2 | 40.0 |
| InstructGPT curie-001 | 0.0 | 0.0 | 26.7 | 16.4 | 40.3 | 0.0 | 5.1 | 51.2 | 0.0 | 4.2 | 55.3 | 46.1 |
| InstructGPT davinci-003 | 0.0 | 0.0 | 14.9 | 12.6 | 42.3 | 6.5 | 39.3 | 63.1 | 11.6 | 26.0 | 76.3 | 59.3 |
| ChatGPT 0613 | 0.0 | 0.2 | 48.4 | 30.8 | 50.4 | 1.7 | 30.8 | 56.7 | 7.1 | 39.3 | 69.7 | 59.3 |
| GPT-4 0613 (June) | 2.7 | 4.5 | 65.9 | 53.5 | 47.6 | 12.7 | 25.9 | 77.5 | 23.1 | 30.6 | 88.6 | 61.0 |
| GPT-4 0613 (October) | 0.9 | 1.4 | 60.9 | 46.0 | 44.4 | 8.0 | 31.7 | 69.0 | 14.8 | 23.2 | 85.1 | 62.8 |
| Flan-T5-XL + CoT | 0.0 | 0.0 | 4.0 | 37.0 | 5.4 | 0.0 | 4.8 | 9.7 | 0.3 | 4.4 | 9.9 | 8.9 |
| Flan-UL2 + CoT | 0.0 | 0.0 | 3.8 | 36.6 | 8.7 | 0.4 | 5.3 | 12.6 | 0.0 | 4.0 | 9.8 | 10.5 |
| Mistral Instruct 7B + CoT | 0.0 | 0.4 | 58.4 | 31.5 | 19.3 | 6.0 | 26.8 | 63.6 | 7.8 | 28.2 | 67.0 | 33.9 |
| Zephyr 7B + CoT | 0.0 | 0.0 | 46.3 | 62.7 | 21.7 | 1.0 | 14.3 | 54.0 | 0.9 | 7.6 | 46.4 | 21.7 |
| Falcon Instruct 7B + CoT | 0.0 | 0.0 | 40.4 | 45.1 | 17.0 | 0.6 | 9.3 | 45.0 | 0.7 | 7.1 | 48.0 | 17.2 |
| Falcon Instruct 40B + CoT | 0.0 | 0.0 | 40.5 | 66.1 | 18.7 | 0.9 | 11.0 | 49.0 | 0.4 | 6.2 | 55.3 | 19.0 |
| Llama-2 Chat 70B + CoT | 0.0 | 0.0 | 53.6 | 28.6 | 20.8 | 2.3 | 20.1 | 56.9 | 4.0 | 21.1 | 63.9 | 32.2 |
| InstructGPT curie-001 + CoT | 0.0 | 0.0 | 10.7 | 16.3 | 39.3 | 0.3 | 6.1 | 53.0 | 0.0 | 2.1 | 50.1 | 38.1 |
| InstructGPT davinci-003 + CoT | 1.2 | 3.0 | 32.8 | 20.6 | 37.2 | 6.5 | 32.5 | 78.9 | 11.6 | 31.4 | 83.7 | 47.1 |
| ChatGPT 0613 + CoT | 0.6 | 1.8 | 52.8 | 47.2 | 41.1 | 11.8 | 34.3 | 70.6 | 15.4 | 32.5 | 74.3 | 51.5 |
| GPT-4 0613 (June) + CoT | 10.1 | 15.4 | 70.1 | 61.2 | 43.9 | 29.6 | 42.0 | 86.1 | 45.6 | 53.3 | 91.0 | 52.2 |
| GPT-4 0613 (October) + CoT | 0.9 | 1.4 | 60.9 | 46.0 | 44.4 | 8.0 | 31.7 | 69.0 | 14.8 | 23.2 | 85.1 | 62.8 |
| InstructGPT curie-001 + FT | 0.0 | 0.0 | 55.1 | 53.7 | 43.6 | 0.7 | 0.7 | 88.3 | 0.1 | 0.0 | 87.3 | 25.4 |
| Flan-T5-XL + FT | 21.3 | 47.1 | 92.0 | 62.4 | 42.6 | 49.3 | 72.8 | 87.2 | 52.2 | 73.5 | 87.2 | 50.2 |

Table 9: Zero-shot results from humans and large language models on 👻 FANToM with the same instructions. CoT denotes chain-of-thought reasoning and FT denotes fine-tuning.