# OpenReview forum: "FANToM: A Benchmark for Stress-testing Machine Theory of Mind in Interactions"
_EMNLP/2023/Conference — EMNLP 2023 Main_

### Official Review · Reviewer_8JV5 · 2023-07-27

**Soundness:** 4

**Excitement:**

4: Strong: This paper deepens the understanding of some phenomenon or lowers the barriers to an existing research direction.

**Paper Topic And Main Contributions:**

The authors propose a benchmark to test Theory of Mind in natural conversations which the authors view as raw interactions as opposed to previous benchmarks consisting of template-based narratives potentially explicitly stating implicit cues. The authors state that FANToM is designed to evaluate a given model’s capability for a coherent understanding of other’s mental states at the example of tracking a model beliefs regarding multiple characters in natural conversations with some info inaccessible to some of these characters. An additional goal is to identify so-called illusory ToM which the authors describe as detecting cases where the model answering some questions correctly, while failing others requiring the same type of ToM reasoning.

**Questions For The Authors:**

- Do you plan to release the benchmark and if so, under which license? What is the timeline for this?
- Abstract: Please state the language that is investigated / benchmark focuses on in the abstract.
- Intro + Related Work: Please add peer-reviewed papers of high quality as citations about the „debate around ToM“ instead of citing 2 New York Times articles only (one behind a paywall…)
- l. 154f: You create „false answers that have high word correlation with the input to verify whether the models can overcome the shortcut pattern matching when reasoning mental states“. I might have missed it in the results/analysis section but as far I saw you only briefly touched upon lexical overlap in 517f in a different context. However, the community might be interested in an analysis of this and potential implications.
- l. 409 + corresponding App: Could you please state how many question sets each of the annotators evaluated with information on whether some worked on substantially more than others? Why did you only ask 1 annotator per question set? Why use AMT and not a different way of evaluation where trust in annotator reliability is a little higher?
- Results Table:
    - Would it be possible to generate a results table where all models’ performances are included (this can go in the appendix, but I find it hard to compare performances across many models with 2 tables).
    - Would it be possible to provide a type of indication of differences in model performance that are statistically significant?
- l. 570, 588, etc.: Can you please elaborate on what is meant with CoT having „a selective impact on different questions types“? Do you have any intuition about why this could be the case?

// edit: Thanks to the authors for their detailed responses.

**Reasons To Accept:**

I like the idea to extend the evaluation of LLMs regarding ToM to conversation-like text. In my opinion, such a benchmark can benefit the research community and is likely to be leveraged be researchers working on ToM in addition to previous benchmarks. I thus think the benchmark is a novel and useful contribution.

Constructing the benchmark using an LLM  is an interesting research question in itself (which, however, also comes with some caveats, see Reasons to Reject) and a potential use case of the benchmark from a meta perspective. It can further be acknowledged that the authors seek to embed their work into the larger theoretical field of ToM and try to address relevant points when developing main design criteria.

The authors provide a first evaluation of their benchmarkd on a range of LLMs including different versions of the same proprietary and open-source models and describe the obtained results in detail. They address main research questions including the identification of potentially illusory ToM according to their definition.

**Reasons To Reject:**

A main reason to reject is the lack of the benchmark itself: How can this be used as a real benchmark if it is not available to anyone? What does „plan to release“ mean? The authors should please state whether they will release the benchmark or not and under which license. As the authors produced FANToM with GPT-based models, it would be helpful if they could elaborate on potential restrictions.
If the the benchmark is not released, the authors should make at least the code for benchmark construction and model evaluation fully available to ensure that other researchers can try to reproduce results (at least on the open-source portion of the LLMs) and potentially build on your work.

A main aspect of the novelty of this work is the evaluation of LLMs regarding ToM in natural conversations vs. other types of conversations / narratives. These conversations are insofar natural as they are not template-based but text obtained by prompting GPT-3.5 text-davinci-003/GPT-4 which are, as far as I understood, in turn also evaluated on the text is generated. This point is a weakness that is in my opinion part of the paradigm that a proprietary, highly under-documented model can be evaluated on a benchmark for the construction of which it was given substantial information about this benchmark. I very much see the point in evaluating on other LLMs, especially open-source models, but unless the authors own that model, I do not think it is very clear what exactly happens with the input. I would have liked to at least see some kind of comment on this.

The construction process is interesting and sound given that one agrees with the paradigm that natural conversations do not need to be generated by a human but can be generated by model. What I am missing regarding the naturalness of conversations is a more nuanced (report of) human validation. The authors state that they conducted a human validation of the benchmark but do not provide any details (e.g., guidelines and validation protocol, how many annotators, what criteria are applied in the recruting process, how well do they agree on which dimensions, etc.). Hence, I am not completely sure about the quality of the benchmark w.r.t. this dimension. Also, it would be helpful if the authors could talk about potential anthropomorphism not only in the ethics section but much earlier to make clear that these conversations are actually a machine-generated version of human mental states. This does not mean that the benchmark is not interesting or of bad quality, but that this should be made known to the reader right in the introduction.

Another point concerns the AMT evaluation yielding the relatively low human upper bound: As far as I understand from the appendix, each question set is evaluated by one annotator. Each annotator could do 1+ question sets. The authors do not report how many question sets each annotator evaluates. Quite some work has shown that AMT annotation might not be the best choice in every scenario due to spammers, unreliable annotators, and this is especially true if, in fact, only one annotator evaluated each questions set.  In case of a potential new benchmark which could be assumed as a substantial extension over previous work, it would be nice to at least have either more annotators per question set and/or one other kind of validation / human performance checkpoint where trust in annotators can be rated higher than for AMT. This could be any kind of non-AMT evaluation, e.g., through offline recruiting, a custom online evaluation, or other service platforms such as Prolific, etc.

**Reproducibility:**

3: Could reproduce the results with some difficulty. The settings of parameters are underspecified or subjectively determined; the training/evaluation data are not widely available.

**Reviewer Confidence:**

3: Pretty sure, but there's a chance I missed something. Although I have a good feel for this area in general, I did not carefully check the paper's details, e.g., the math, experimental design, or novelty.

---

> ### Author Rebuttal · Authors · 2023-08-28
>
> We thank you for the appreciation of our benchmark to be “a novel and useful contribution” and our efforts for “embedding our work into the larger theoretical field of ToM”!
>
> **1. Release of dataset and codes:**
> Our dataset and codes are currently fully prepared to be publicly released upon acceptance, under CC-BY license. We are enthusiastic to share our dataset and code, so that other researchers can reproduce and build on our work!
>
> **2. Evaluating a closed, proprietary model on the benchmark that was generated from it:**
> Thank you for that point! In light of the theme of this year’s EMNLP, we evaluate not only various open- and closed-source models but also GPT-4, to show that even the model that was used to generate the dataset is unable to solve it. Also, although it is a closed model, it is often seen as the most capable model and can thus serve as an “upper bound or existential proof that can be achieved with some unspecified combination of current methods and data” [1]. We will make sure to include the discussion around this issue in the limitations section of the updated draft.
>
> **3. Human validation and the quality of generated conversations:**
> We conducted an additional round of validation on the entire conversations in our dataset using 32 annotators who passed a qualification test for assessing conversation coherency on Amazon Mechanical Turk. We asked workers to flag conversations that were incoherent or unsafe (e.g., unethical, biased, harmful, dangerous, or offensive). Each conversation was validated by three workers. While 10 conversations received votes for incoherency, none achieved a majority vote indicating they were incoherent. This is in line with previous works showing machine-generated conversations to have great quality [2, 3]. As for safety, no conversations were voted as being unsafe.
>
>
> **4. Moving up discussions of anthropomorphism:**
> Good point! We fully agree with you that it is important to clear any potential misunderstanding regarding anthropomorphism! We will move up the discussions about potential anthropomorphism to the introduction section, including the nature of the conversations in our benchmark. We will also indicate that these are "machine-generated" conversations, "simulating" human interactions, starting from the abstract.
>
> **5. Measuring human performance with workers from Amazon Mechanical Turk (AMT):**
> Unfortunately, AMT was our best option, as our institution has prohibited the use of Prolific due to data concerns. We would like to emphasize that we sought to have an identical setup to the models without introducing any separate qualifications or tutorials. Nevertheless, we recognize that this might have introduced some noise in measuring human performance. In our updated draft, we will set an additional human performance baseline by recruiting offline annotators (e.g., graduate students at our institution). However, we would like to gently remind that even the recent best performing models are still far below the current human baseline. For the present baseline, each worker solved an average of 12 question sets.
>
> **6. The selective impact of chain-of-thought (CoT) reasoning:**
> [L570] Results in Figure 2 & 3 show that CoT reduces false positive errors (i.e., saying yes for unaware characters for binary questions, and including them in the response for list type questions), while it increases false negative errors (e.g., saying no for characters who are aware of the info for binary questions, and excluding them in the response for list type questions).
>
> [L588] CoT decreases the likelihood of correctly answering all questions related to a specific character (i.e., 'All for Each Character' in Table 2) for some models, while improving the overall performance shown in Table 1. We will elaborate this in the updated draft.
>
> **7. Other comments:**
> Thank you for the suggestions! We will reflect all your comments: mentioning base language, benchmark focus, additionally citing articles not behind paywalls, updating a full table with all model results in appendix, and adding statistical significance.
>
> [1] Anna Rogers 2023 [Closed AI Models Make Bad Baselines - Hacking semantics](https://towardsdatascience.com/closed-ai-models-make-bad-baselines-4bf6e47c9e6a)
>
> [2] Chen, Maximillian, et al. "PLACES: Prompting Language Models for Social Conversation Synthesis." Findings of the Association for Computational Linguistics: EACL 2023. 2023.
>
> [3] Kim, Hyunwoo, et al. "Soda: Million-scale dialogue distillation with social commonsense contextualization." arXiv preprint arXiv:2212.10465 (2022).

---

### Official Review · Reviewer_iirp · 2023-08-04

**Soundness:** 4

**Excitement:**

2: Mediocre: This paper makes marginal contributions (vs non-contemporaneous work), so I would rather not see it in the conference.

**Paper Topic And Main Contributions:**

The paper presents a new benchmark for testing theory of mind in multi-party conversations with asymmetric information. Specifically, in conversations, the benchmark tests the belief of characters (what does X believe), the answerability of questions (who knows the right answer to X), and which characters possess information (who has information on X). The dataset is comprised of 256 conversations with ~6k questions total, a mixture of the aforementioned types. Evaluation of existing models shows that they perform significantly worse compared to human performance.

**Questions For The Authors:**

InfoAccess and Answerability seem to be very similar? As in, I would expect that a character who knows a given piece of information implies that it would know the correct answer to the question? Not sure what exactly the difference is here.

**Reasons To Accept:**

An interesting new asymmetric-information motivated theory of mind benchmark in the multiparty conversation setting. The authors source from key ToM principles in the construction and design of their dataset, and existing models are shown to perform badly on this even with significant chain of thought prompting.

**Reasons To Reject:**

I'm curious to see how this dataset, or its principles, can be used to make existing models better. It works as, and *only* as, an evaluation of the capabilities of existing models, but the paper leaves a lot to be desired on how we can build systems that can possess better ToM capabilities. Some experimental design on this front would have significantly improved the impact of the paper.

**Reproducibility:**

5: Could easily reproduce the results.

**Reviewer Confidence:**

3: Pretty sure, but there's a chance I missed something. Although I have a good feel for this area in general, I did not carefully check the paper's details, e.g., the math, experimental design, or novelty.

---

> ### Author Rebuttal · Authors · 2023-08-28
>
> We appreciate your positive acknowledgement of our work as a new interesting benchmark, as well as your recognition of our benchmark design with key ToM principles!
>
> **1. How this dataset, or its principles, can be used to make existing models better:**
> While we have discussed that applying chain-of-thought reasoning and fine-tuning can improve the vanilla performance, we will also discuss potential future directions for researchers to tackle this challenging benchmark better. For example, tracking mental states with belief graphs [1], structured prediction between question types to ensure consistency, and procedural prompting in the order of the easiest questions to the most difficult questions (e.g., InfoAccessibility questions to Answerability questions to Belief questions).
>
> Also, we would like to gently note that the principles of our benchmark (e.g., information asymmetry, minimizing reporting bias, theoretical requirements) can be crucial for clarifying the recent illusion around the ToM capabilities of AI models. These principles also introduce underlying key components for social reasoning, such as understanding information access and asymmetry. As efforts increase in evaluating social reasoning of LLMs, it is important to establish elaborate benchmark designs early on. Thankfully, other reviewers such as 8JV5 and we1R have acknowledged this importance too. We believe our work can stimulate meaningful discussions within the research community.
>
>
> **2. Difference between InfoAccessibility and Answerability:**
> Thank you for bringing up this point! You are right in saying that “if a character who knows a given piece of information implies that it would know the correct answer to the question.” This is a very obvious reasoning step for us humans. However, as we show in Table 1, this does not hold true for models, as the performance difference between the InfoAccessibility and Answerability question is significant. We designed these two questions precisely to tease out whether the models can follow this internally consistent reasoning process. We will expand on this important point in the discussion of the revised draft.
>
> To clarify, the key difference between InfoAccessibility and Answerability questions lies in the depth of reasoning required. The InfoAccessibility question simply requires to reason who knows the given information piece (i.e., one-step reasoning). In contrast, the Answerability question requires (1) determining the answer of the question, and (2) reasoning which character has access to this information piece (i.e., two-step reasoning).
>
> [1] Melanie Sclar, et al. 2023. Minding Language Models’ (Lack of) Theory of Mind: A Plug-and-Play Multi-Character Belief Tracker. In Proceedings of the 61st Annual Meeting of the Association for Computational Linguistics (Volume 1: Long Papers), pages 13960–13980

---

### Official Review · Reviewer_o4fC · 2023-08-09

**Soundness:** 3

**Excitement:**

3: Ambivalent: It has merits (e.g., it reports state-of-the-art results, the idea is nice), but there are key weaknesses (e.g., it describes incremental work), and it can significantly benefit from another round of revision. However, I won't object to accepting it if my co-reviewers champion it.

**Paper Topic And Main Contributions:**

This paper introduces a dataset, FanToM, to benchmark model performance on Theory of Mind (ToM).  It provides empirical results of language models on the dataset.

**Reasons To Accept:**

Benchmark of language models on ToM.

**Reasons To Reject:**

N/A

**Reproducibility:**

4: Could mostly reproduce the results, but there may be some variation because of sample variance or minor variations in their interpretation of the protocol or method.

**Reviewer Confidence:**

1: Not my area, or paper was hard for me to understand. My evaluation is just an educated guess.

---

> ### Author Rebuttal · Authors · 2023-08-28
>
> Thank you for the positive endorsement!

---

### Official Review · Reviewer_we1R · 2023-08-12

**Soundness:** 3

**Excitement:**

4: Strong: This paper deepens the understanding of some phenomenon or lowers the barriers to an existing research direction.

**Paper Topic And Main Contributions:**

The authors present FANToM, a benchmark for analyzing theory-of-mind capabilities of language models in conversation settings. This setting primarily focuses on information-asymmetric conversations where some individuals have partial information. They construct a benchmark using GPT-3.5 to create samples based on a set of rules, and perform a number of validations. They then evaluate the benchmark on several LLMs, from which they find that LLMs significantly underperform humans in the tasks, and that chain-of-thought reasoning improves performance but only to a certain extent in certain scenarios.

**Questions For The Authors:**

- I am curious why the authors chose the current task setting of having participants explicitly leave and rejoin during conversations to measure information asymmetry. What difference does it have compared to say, masking some parts of the conversation when inputting into a model?
- What is the total cost of constructing the dataset?


**Reasons To Accept:**

- This paper proposes a benchmark for evaluating the theory-of-mind in conversation settings, which can be of large importance to the LLM research community that will be interested in studying its capabilities in conversations
- The benchmark is well-designed, both in terms of underlying theory and coverage
- The evaluation is conducted on a large set of LLMs and are able to provide insights on their capabilities in conversation settings

**Reasons To Reject:**

- The emphasis on information-asymmetric conversation settings feels a bit arbitrary and is not justified well enough.

**Reproducibility:**

4: Could mostly reproduce the results, but there may be some variation because of sample variance or minor variations in their interpretation of the protocol or method.

**Reviewer Confidence:**

2: Willing to defend my evaluation, but it is fairly likely that I missed some details, didn't understand some central points, or can't be sure about the novelty of the work.

**Typos Grammar Style And Presentation Improvements:**

- The abstract could be a bit more informative

---

> ### Author Rebuttal · Authors · 2023-08-28
>
> Thank you for the positive feedback that our benchmark is well designed and that it “can be a large importance to the LLM research community interested in studying its capabilities in conversation”!
>
> **1. The justification of the emphasis on information asymmetry:**
> Thank you for bringing up this question! In psychology, many existing theory of mind tasks can be viewed as having some form of asymmetry between characters [1]. For example, in the Sally-Anne task, Sally does not know that Anne relocated the object, while the observer is aware of the action. In the Smarties task, the character in the story does not know the label changed, whereas the observer is fully aware of this situation. This inherent asymmetry ensures two distinct mental states to be present during the experiments. Therefore, information asymmetry is a key component when designing a theory of mind benchmark. We will clarify this in the updated draft.
>
> **2. The purpose of having participants explicitly leave and rejoin the conversation:**
> Thank you for asking! By explicitly surfacing the absence of the character, it becomes clearer for the observer that the specific character does not have access to certain information. Many existing theory of mind tasks in psychology also include some form of explicit indication of absence (e.g., Sally-Anne task). This ensures a clear distinction in the mental states between the character and the observer (i.e.,  fulfilling the non-merging criterion in Section 2), while maintaining a natural and self-contained context. Without such explicit indication, the character's presence or absence becomes ambiguous, especially since our benchmark relies solely on text.
>
> Masking certain areas of the text violates the non-merging criterion which is the basic requirement of our task: if the part where the certain character was absent is masked when given to the model, the model will also lack access to the information, eliminating the information asymmetry.
>
> **3. Total cost of constructing the benchmark:**
> It was less than 1000 USD.
>
> **4. More information needed in abstract:**
> We will make sure to update the abstract to be more informative.
>
> [1] Braüner, Torben, Patrick Blackburn, and Irina Polyanskaya. "Being Deceived: Information Asymmetry in Second‐Order False Belief Tasks." Topics in Cognitive Science 12.2 (2020): 504-534.

---

### Meta-Review · Area_Chair_28Hd · 2023-09-18

**Recommendation:** 4

**Metareview:**

The paper proposes a benchmark for theory of mind conversations, which is novel and timely wrt to the current pace of LLM development. Authors in their rebuttal also confirm the intention to release this benchmark for general use ASAP, which is the most valuable part. Contingent on this release, the paper can be very helpful for LLM evaluations. I would also urge authors to release a datasheet, detailing aspects like cost of data construction, intended usage, etc. This recurs in reviewers’ questions and underlines how any reader or user will have the same questions.
Additionally, like reviewers point out, some of the discussions, specifically around anthropomorphism and limitations wrt it should appear much earlier in the paper to establish the context of benchmark usage.
With these updates, the work would be of great use to the community.

---

### Decision · Program_Chairs · 2023-10-07

**Decision:**

Accept-Main

**Comment:**

The paper proposes a benchmark for theory of mind conversations, which is novel and timely wrt to the current pace of LLM development. Authors in their rebuttal also confirm the intention to release this benchmark for general use ASAP, which is the most valuable part. Contingent on this release, the paper can be very helpful for LLM evaluations. I would also urge authors to release a datasheet, detailing aspects like cost of data construction, intended usage, etc. This recurs in reviewers’ questions and underlines how any reader or user will have the same questions.
Additionally, like reviewers point out, some of the discussions, specifically around anthropomorphism and limitations wrt it should appear much earlier in the paper to establish the context of benchmark usage.
With these updates, the work would be of great use to the community.